# CircAgtpbp1 Acts as a Molecular Sponge of miR-543-5p to Regulate the Secretion of GH in Rat Pituitary Cells

**DOI:** 10.3390/ani11020558

**Published:** 2021-02-20

**Authors:** ZeWen Yu, WenZhi Ren, Tian Wang, WeiDi Zhang, ChangJiang Wang, HaoQi Wang, Fei Gao, Bao Yuan

**Affiliations:** Department of Laboratory Animals, College of Animal Sciences, Jilin University, 5333 Xi’an Road, Changchun 130062, China; yzw19950219@163.com (Z.Y.); renwz@jlu.edu.cn (W.R.); wangtian19861023@sina.com (T.W.); zhangwd19@mails.jlu.edu.cn (W.Z.); wangcj1105@163.com (C.W.); hqwang1997@163.com (H.W.)

**Keywords:** circRNA, miRNA sponge, GH, pituitary, animal growth

## Abstract

**Simple Summary:**

Circular RNAs(circRNAs) and microRNAs(miRNAs) are a type of endogenous non-coding RNA that are widely expressed in tissues and play an important role in growth and development. Growth hormone (GH) regulates the growth and apoptosis of cells. However, there are still few reports on the mechanisms involving circRNA and GH. Therefore, this study aimed to explore how circRNAs regulate the secretion of GH. We studied a circRNA named circAgtpbp1 from our previous RNA sequencing, which can target miR-543-5p binding to the 3’UTR of Gh1 messager RNA (mRNA). Our results demonstrate that circAgtpbp1 is a stable, truly circular molecule, which is located on chromosome 17. Moreover, we found that circAgtpbp1 can promote the secretion of GH by attenuating the effect of miR-543-5p. These findings expand our existing knowledge of the mechanisms of hormone regulation and enrich the research involving the regulation of hormones by non-coding RNA.

**Abstract:**

CircRNAs have been identified to be expressed differently and stably in numerous species and tissues, but their functions in growth hormone (GH) secretion are still largely unknown. In summary, we have revealed a circRNA-miRNA-mRNA network that may play a biological role in the rat pituitary gland. First, we verified the chromosome location information of circAgtpbp1 according to sequencing analysis. The circAgtpbp1 characteristics were authenticated through PCR, qRT–PCR, treating with RNase and fluorescent in situ hybridization (FISH). Second, we detected the expression pattern of circAgtpbp1 in the rat anterior pituitary by qRT–PCR. We also designed circAgtpbp1 siRNA and constructed overexpression plasmid to evaluate the effect of circAgtpbp1 function on GH secretion by qRT–PCR, ELISA and Western blot. CircAgtpbp1 is a stable, truly circular molecule. We found that circAgtpbp1 interacted with miR-543-5p and can regulate GH secretion in pituitary cells through a circAgtpbp1-miR-543-5p-GH axis. Overall, the evidence generated by our study suggests that circAgtpbp1 can act as a sponge of miR-543-5p to reduce the inhibitory effect of miR-543-5p on Gh1 and further promote GH secretion. These findings expand our existing knowledge on the mechanisms of hormone regulation in the pituitary gland.

## 1. Introduction

The pituitary gland is the most important endocrine organ and is heterogeneous among mammals [1]. The pituitary is composed of the posterior pituitary and anterior pituitary, which are also called the neurohypophysis and the adenohypophysis, respectively [2]. There are a variety of pituitary cell types (from somatotroph, lactotroph, thyrotroph, corticotroph and gonadotroph cell lineages) [3]. All types of pituitary cells are regulated by hypothalamic factors and secrete specific hormones [4]. Growth hormone (GH) is one of the hormones secreted by the somatotroph cell lineage.

GH is a polypeptide hormone that is essential for somatic cells [5]. It regulates many growth-promoting, anabolic and lipolytic processes [6,7,8,9]. GH secretion dysfunction leads to specific diseases. For example, excessive GH secretion causes gigantism and acromegaly [10,11]. Excessive GH also affects aging and certain cancers [12,13]. In contrast, insufficient GH secretion causes dwarfism [14]. These effects imply that maintenance of GH at stable levels suitable for physiological homeostasis is important [15]. The process of GH synthesis is mainly controlled by the hypothalamic neuropeptides GH-releasing hormone (GH-RH) and somatotropin release–inhibiting factor (SRIF; also known as somatostatin) [16,17] In addition to these two classic regulatory factors, various neurotransmitters, neurohormones and peripheral factors have been reported to play roles in GH secretion [18,19]. Thus, an in-depth understanding of the molecular mechanism of GH regulation is crucial.

Circular RNAs (circRNAs) are novel endogenous noncoding RNA molecules composed of unique covalently closed continuous loops without 5′ caps or 3′ poly(A) tails [20]. Unlike linear RNA molecules, circRNAs are more stable after treatment with RNase R because of their loop structures [21]. Most circRNAs originate from exons, and less are from introns; both types are produced through back-splicing [22,23]. Many studies have suggested that circRNAs are present in various cells and tissues [24,25,26,27] and are involved in diverse malignant behaviors, such as cancer cell migration [28], invasion [29], apoptosis [30] and proliferation [31]. CircRNAs can also affect many biological processes under normal physiological conditions [32,33]. In our previous study, we identified a circRNA that can act as a miRNA sponge to regulate the FSHβ gene when circRNA functions as competing for endogenous RNAs(ceRNA), which could regulate gene expressions through binding to miRNA; we call this mechanism as ceRNA network [34]. However, much is still unknown about the roles of ceRNA networks in regulating animal growth and development through Gh1. In this study, we identified the novel circRNA rno_circ_0001059, which we named circAgtpbp1. Furthermore, we found that circAgtpbp1 acted as a molecular sponge of oncogenic miR-543-5p to reduce the inhibitory effect of miR-543-5p on the Gh1 gene by RT–qPCR, ELISA, Western blot analysis, RNA immunoprecipitation (RIP), fluorescence in situ hybridization (FISH) and luciferase reporter functional assays. In summary, we have revealed a circRNA-miRNA-mRNA network that may play a biological role in the rat pituitary gland.

## 2. Materials and Methods

### 2.1. Ethics Statement

All experiments were performed in accordance with the relevant guidelines of the Guide for the Care and Use of Laboratory Animals of Jilin University. All Sprague-Dawley (SD) rats used in this study were obtained from Liaoning Changsheng Biotechnology Co. and given food and water at the Jilin Provincial Key Laboratory of Animal Models. All animal procedures were conducted according to a protocol approved by the Institutional Animal Care and Use Committee (IACUC) of Jilin University (permit number: SY201906012).

### 2.2. Animals and Cell Culture

SD rats with a production license and quality certificate were purchased from Liaoning Changsheng Biotechnology Co. Rat anterior pituitary cells were obtained from 4-month-old male SD rats for primary cell culture. The method used to obtain and culture the rat anterior pituitary cells has been described in our previous study. We removed the heads of rats first and sheared the skin between the ears and mouth. Then opened the skull with tweezers, removed the pituitary glands and placed them into precooled PBS supplemented with 0.3% BSA (Sigma, St. Louis, MO, USA) and 1% penicillin/streptomycin (HyClone, Logan, UT, USA) to wash off the blood from the pituitary. Separating the neurohypophysis from the pituitary and cut the pituitary into pieces in Dulbecco’s modified Eagle’s medium/nutrient mixture F12 (DMEM/F12) (HyClone, USA), containing 2.5% collagenase type I (Gibco, Grand Island, NY, USA). After a 90 min incubation with atmosphere 5% CO_2_ at a temperature of 37 °C, we diluted the pituitary cells with PBS (0.3% BSA and 0.1% penicillin/streptomycin) and then filtered the mixture. Next, the cell suspension was centrifuged at 200× *g* for 10 min. Finally, 2 mL of the DMEM/F12 culture medium with 20% fetal bovine serum (FBS) was used to resuspend the cell pellet [35]. The rat pituitary adenoma cell lines GH3 and MMQ were obtained from the National Infrastructure of Cell Line Resource (resource numbers: 3111C0001CCC000008, 3111C0001CCC000081). All adenoma cell lines were grown in Dulbecco’s modified Eagle’s medium (DMEM)/nutrient mixture F12 (F12) (HyClone) supplemented with 10% fetal bovine serum (FBS) (Gibco, USA) and 100 U/mL penicillin/streptomycin at 37 °C in an atmosphere of 5% CO_2_.

### 2.3. Short Interfering RNA (siRNA) Synthesis and Plasmid Construction

CircAgtpbp1 siRNAs and their negative controls were designed and synthesized by RiboBio Biotech Co., Ltd. (Guangzhou, China). The siRNAs were designed to specifically target the back-spliced junction point so that they would not knock down the expression level of Agtpbp1 mRNA. A circAgtpbp1 expression plasmid was constructed by inserting the whole sequence of circAgtpbp1 into pCD2.1-ciR (Wuhan GeneCreate Biological Engineering Co., Ltd., Wuhan, China). A sequence from approximately 200 base pairs upstream to 200 base pairs downstream of the circAgtpbp1 binding site for miR-543-5p were cloned into the pmirGLO plasmid to obtain pmirGLO-circAgtpbp1-WT. Then, a sequence in which the circAgtpbp1 binding site for miR-543-5p was mutated was used to obtain the pmirGLO-circAgtpbp1-MUT plasmid.

### 2.4. Cell Transfection

Anterior pituitary cells (3 × 10^5^) were seeded in a 24-well plate, and GH3 cells (5 × 10^5^) were seeded in a 12-well plate. All transfection experiments were performed with a lipofectamine 2000 transfection kit (Thermo Fisher Scientific, Waltham, MA, USA) following the manufacturer’s recommended protocols. The final concentrations of the miRNA negative control, miRNA mimic, circAgtpbp1 siRNA negative control, and circAgtpbp1 siRNA were 100 nM, while the final concentrations of the circAgtpbp1 plasmids were 500 ng/well. The cells were harvested and used 24 h after transfection.

### 2.5. RNA Isolation, RNase R Treatment and RT–qPCR

Total RNA was extracted from cells or tissues using TRIzol reagent (Invitrogen, Carlsbad, CA, USA) according to the manufacturer’s protocol. For RNase R treatment, 1 μg of total RNA was incubated for 30 min at 37 °C with or without 3 U/μg of RNase R (Epicentre Technologies, Madison, WI, USA) in 1× RNase R reaction buffer. A FastQuant RT Reagent Kit with gDNA Wiper (Tiangen Biotech Co., Ltd., Beijing, China) was used to synthesize complementary DNA (cDNA). An E.Z.N.A. tissue DNA kit (OMEGA Bio-Tek Inc., Norcross, GA, USA) was used to synthesize genomic DNA (gDNA). The cDNA and gDNA PCR amplification products were observed by 2% agarose gel electrophoresis using a 2× Taq PCR Master Mix (Tiangen Biotech Co., Ltd.) following the manufacturer’s instructions. PCR product sequencing was performed by Comate Bioscience Co., Ltd. RT–qPCR was performed using a SuperReal PreMix Plus Kit (SYBR Green). Quantitative RT–PCR was subsequently performed with Mastercycler ep Realplex^2^ system (Eppendorf, Germany) according to the manufacturer’s instructions (Tiangen Biotech Co., Ltd.). Each well needed 10 μL 2× SuperReal PreMix Plus, 0.5 μL forward primer, 0.5 μL reverse primer, 1 μL cDNA and 8 μL RNase-free ddH_2_O The procedure included initial denaturation at 95 °C for 15 min, followed by 40 cycles 10 s at 95 °C and 30 s at 60 °C. Before analysis by the 2−ΔΔCt method, GAPDH levels were used as a reference to normalize the circRNA and mRNA levels, while U6 snRNA levels were used to normalize the miRNA levels. The sequence information for the primers is listed in Appendix A.

### 2.6. Identification of Target Genes by Target Prediction

Two target prediction tools, RNAhybrid 2.2 [36] and RNA22 v2 [37] were used to screen circRNAs with binding sites for the target miRNA. These two tools use different algorithms. Therefore, combining their results can improve the accuracy of prediction. The circRNA identified in this study was predicted by both RNAhybrid 2.2 and RNA22 v2.

### 2.7. RIP Assay

RIP assays were performed using a Magna RIP RNA-binding protein immunoprecipitation kit (Millipore, Billerica, MA, USA) according to the manufacturer’s protocol. Briefly, cells were lysed using RIP lysis buffer containing protease and RNase inhibitors (Millipore). Then, the cell lysates were incubated with an anti-AGO2 antibody or a nonspecific IgG antibody (Abcam, Cambridge, MA, USA) at 4 °C overnight. The immunoprecipitated RNA was eluted with proteinase K, and circAgtpbp1 enrichment was detected by qRT–PCR.

### 2.8. FISH

Cy3-labeled probes specific for circAgtpbp1 and Fam-labeled miR-543-5p probes were designed and synthesized by Ribo Biotech Co., Ltd. (Guangzhou, China). The anterior pituitary cells were washed in PBS for 5 min and then fixed in 4% paraformaldehyde for 10 min. Using PBS, we washed the cells three times, each time for 5 min. We added 1 mL precooled permeabilizer to each well. We then kept the cells at 4 °C for 5 min. After discarding the permeabilizer, we washed the cells with PBS again. Then, we added 200 µL of prehybridization solution to each well and incubated cells at 37 °C for 30 min. The hybridization solution was preheated at 37 °C. The FISH probe mix stock solution and internal reference FISH probe mix stock solution were added to the hybridization solution. We discarded the prehybridization solution and added 100 μL of probe-containing hybridization solution. Then, we washed the cells three times with wash solution I (4× saline sodium citrate (SSC) and 0.1% Tween-20) for 5 min per wash, once with wash solution II (2× SSC) and once with wash solution III (1× SSC). Then, the cells were washed with PBS for 5 min at room temperature. The cells were stained with DAPI for 10 min and then washed 3 more times with PBS for 5 min each time. FISH was used to determine the localization of circAgtpbp1 and miR-543-5p. Fluorescence microscopy (Olympus, Tokyo, Japan) was used to capture the images.

### 2.9. GH Detection

After transfecting cells with circAgtpbp1 siRNA and circAgtpbp1 overexpression plasmids for 24 h, we cultured the cells with a serum-free medium instead of DMEM-F12 with FBS. After 24 h, we collected the culture medium and measured the levels of secreted GH with a Rat GH ELISA Kit (Shanghai Enzyme-linked Biotechnology Co., Ltd., Shanghai, China).

### 2.10. Western Blot Analysis

After transfection for 48 h, cells were harvested and placed in a cell culture plate. Then, RIPA protein lysis buffer with 1% PMSF was used to lyse the cells. The protein concentrations were determined by using a BCA assay kit (Millipore, Billerica, MA, USA).

The total protein concentration in each of the spleen samples was determined using a BCA protein assay kit. The BCA working reagent was prepared by mixing BCA reagent A with BCA reagent B at a ratio of 50:1. In addition to BCA reagent B to A, turbidity was observed, which quickly disappeared upon mixing, yielding a clear green working reagent. About 200 µL of BCA working reagent was added into each well containing standards and samples at the ratio of 1:8. The plate was covered and incubated at 37 °C for 30 min. The plate was cooled down to room temperature, and the absorbance was measured at 562 nm on a plate reader. Blank-corrected absorbance of each individual standard and sample replicate was calculated by subtracting the average absorbance measurement of the blank sample from the absorbance measurements of each individual standard and sample replicates. A standard curve was plotted using Excel and was used to determine the protein concentration of each of the unknown samples. Next, the proteins in the supernatant were transferred to PVDF membranes for 1 h. The membranes were incubated in 50 mL of 5% nonfat dry milk in TBST for 1 h at room temperature before being incubated with specific primary antibodies (anti-β-actin and anti-Gh1) at 4 °C overnight. The next day, the membranes were washed three times for 10 min each in TBST and incubated with secondary antibodies for 1 h at room temperature. MilliporeSigma™ Immobilon™ Western Chemiluminescent HRP Substrate (ECL) (WBKL0500, Millipore) was used to detect the protein bands, and the ImageJ program was used to perform protein quantification.

## 3. Results

### 3.1. Identification and Validation of the circRNA

In a previous study, we found that miR-543-5p can bind to the 3’UTR of Gh1 mRNA [35]. To identify the other regulatory factors involved in the molecular mechanism of GH secretion, we searched for circRNAs targeting miR-543-5p in our previous RNA sequencing results for D15 and D120, which included a total of 4123 circRNAs [38]. Through miRNA target prediction with RNAhybrid 2.2 and RNA22 v2, rno_circ_0001059 was identified as having base pairs that match miR-543-5p (Figure 1A). Gel electrophoresis verified the PCR product of the screened circRNA (Figure 1B), and Sanger sequencing experiments confirmed that rno_circ_0001059 is composed of 14–19 exons of the ATP/GTP binding protein 1 (Agtpbp1) gene, whose mature spliced sequence length is 1227 bp. The full sequence of circAgtpbp1 is provided in Appendix A. This gene is located on chromosome 17:5,555,491–5,571,375, herein termed circAgtpbp1. Sequencing of the PCR product across the splice site demonstrated the presence of circAgtpbp1 (Figure 1C). Next, we designed convergent primers to amplify Agtpbp1 mRNA and divergent primers to amplify circAgtpbp1. Agtpbp1 mRNA was amplified from both gDNA and cDNA templates from rat pituitary cells by the convergent primers, but circAgtpbp1 was amplified only from cDNA by the divergent primers (Figure 1D). After treatment with RNase R, the expression level of circAgtpbp1 was significantly higher than the expression level of Agtpbp1 mRNA because circRNAs with closed-loop structures are more stable than linear mRNA molecules. Thus, circAgtpbp1 remained stable, but linear Agtpbp1 was almost entirely degraded (Figure 1E). Furthermore, random hexamer and oligo (dT)_18_ primers were used for reverse transcription, and we found that the relative expression of circAgtpbp1 reverse transcribed with the random hexamers was significantly higher than that of circAgtpbp1 reverse transcribed using the oligo (dT)_18_ primers (Figure 1F). FISH against circAgtpbp1 demonstrated that circAgtpbp1 was mainly localized in the cytoplasm (Figure 1G).

### 3.2. CircAgtpbp1 Expression Pattern in the Rat Anterior Pituitary

To determine the expression pattern of circAgtpbp1 in the rat anterior pituitary, we detected the expression level of circAgtpbp1 in anterior pituitary samples from rats at different ages: 7 days, 40 days, 90 days and 250 days. The results showed that 7-day-old rats exhibited significantly higher expression levels of circAgtpbp1 than rats of other ages (Figure 2A). The expression levels of circAgtpbp1 in different tissues of rats, including the pituitary, liver, spleen, lungs, kidneys, hindbrain, and heart, were obviously different and were significantly higher in the pituitary gland than in other tissues (Figure 2B). RT–qPCR analysis of circAgtpbp1 expression levels in different cells showed that circAgtpbp1 levels were the same in GH3 cells and MMQ cells (Figure 2C). Therefore, we chose GH3 cells for experiments because GH3 cells are adherent and are easy to transfect.

### 3.3. Effects of CircAgtpbp1 Knockdown on Gh1 Transcription

To explore the functional role of circAgtpbp1 in GH secretion in the rat pituitary, we designed three siRNAs that specifically targeted the back-spliced junction point of circAgtpbp1 and transfected them into GH3 cells and rat pituitary cells to achieve circAgtpbp1 knockdown (Figure 3A). RT–qPCR assays revealed that all three siRNAs could stably downregulate the expression of circAgtpbp1, and siRNA-2 was chosen for subsequent experiments (Figure 3B,C). RT–qPCR assays and Western blotting experiments revealed that Gh1 mRNA levels and Gh1 protein levels were significantly lower in cells transfected with siRNA-2 than in negative control cells (Figure 3D–G). ELISA showed that silencing circAgtpbp1 significantly suppressed GH secretion (Figure 3H,I). These results demonstrate that the knockdown of circAgtpbp1 suppresses Gh1 mRNA expression and GH secretion.

### 3.4. Effects of CircAgtpbp1 Overexpression on Gh1 Transcription

To further examine the role of circAgtpbp1 in the regulatory mechanism of GH secretion, the circAgtpbp1 overexpression vector pCD2.1-circAgtpbp1 or the empty plasmid pCD2.1 was transfected into GH3 cells and primary rat pituitary cells. RT–qPCR revealed that the expression level of circAgtpbp1 was significantly increased by the overexpression vector (Figure 4A,B). In addition, the *Gh1* expression level was higher in the overexpression group than in the negative control group (Figure 4C,D), as was the Gh1 protein level (Figure 4E,F). ELISA revealed that GH secretion was promoted by circAgtpbp1 overexpression (Figure 4G,H). Taken together, these results suggest that circAgtpbp1 plays a role as a regulator of GH secretion in rat pituitary cells.

### 3.5. CircAgtpbp1 Sponges miR-543-5p in Primary Rat Pituitary Cells

Since circRNAs can function as miRNA sponges to inhibit miRNA expression, we sought to verify whether circAgtpbp1 could act as a sponge of miR-543-5p in primary rat pituitary cells. Thus, we constructed a pmirGLO plasmid containing a sequence from approximately 200 bases upstream of the circAgtpbp1 binding site to approximately 200 bases downstream of the binding site (Appendix A). We also mutated the binding site. Ultimately, we obtained pmirGLO-circAgtpbp1-wt and pmirGLO-circAgtpbp1-mut plasmids. In cells cotransfected with the miR-543-5p mimic and pmirGLO-circAgtpbp1-wt, the luciferase activity of pmirGLO-circAgtpbp1-wt was dramatically inhibited by miR-543-5p. However, luciferase activity was restored by mutation of the predicted miR-543-5p binding site (Figure 5A). To further confirm the binding of these molecules, we performed a RIP assay with an AGO2 antibody. After overexpression of miR-543-5p, circAgtpbp1 was greatly enriched, indicating substantial accumulation of bound circAgtpbp1 and miR-543-5p in the AGO2 pellet (Figure 5B). An RT–qPCR assay suggested that knockdown of circAgtpbp1 expression promoted miR-543-5p expression in rat pituitary cells and GH3 cells (Figure 5C,D). In contrast, overexpression of circAgtpbp1 markedly suppressed miR-543-5p expression (Figure 5E,F). Details on the construction of the pmirGLO-circAgtpbp1-wt and pmirGLO-circAgtpbp1-mut reporter plasmids are provided in Appendix A. Taken together, these results show that circAgtpbp1 promotes GH secretion by acting as a sponge of miR-543-5p.

### 3.6. CircAgtpbp1 Regulates Gh1 Expression and GH Secretion by Targeting miR-543-5p

To clarify the function of circAgtpbp1/miR-543-5p/Gh1 in the rat pituitary, rescue experiments were performed. First, GH3 rat pituitary cells were cotransfected with pCD2.1 and a miR-543-5p negative control, si-circAgtpbp1 and a miR-543-5p negative control, or si-circAgtpbp1 and a miR-543-5p inhibitor. The RT–qPCR results revealed that knockdown of circAgtpbp1 suppressed *Gh1* mRNA expression (Figure 6A). However, the suppression of *Gh1* mRNA was partly attenuated in cells cotransfected with si-circAgtpbp1 and the miR-543-5p inhibitor (Figure 6A). In contrast, the relative expression level of Gh1 was significantly increased after overexpression of circAgtpbp1, while cotransfection with the pCD2.1-circAgtpbp1 plasmid and the miR-543-5p mimic counteracted this increase (Figure 6C). Gh1 mRNA levels exhibited the same trends in rat pituitary cells after transfection (Figure 6B,D). These results indicate that circAgtpbp1 can promote Gh1 expression via the circAgtpbp1/miR-543-5p/Gh1 axis.

## 4. Discussion

CircRNAs are a group of unique noncoding RNAs (ncRNAs) that have a covalently closed loop structure [39] and are generated by back-splicing of upstream 3′ splice sites and downstream 5′ splice sites [40]. According to their different compositions, circRNAs can be classified as exonic circular RNAs (ecRNAs), which are present in the cytoplasm, or as intronic circRNAs (ciRNAs) or exon–intron circRNAs (EIciRNAs), which are present in the nucleus [40,41]. CircRNAs that play roles as molecular sponges of miRNAs are generated from exons [42,43,44]. In our previous study, we found that miR-543-5p can downregulate the expression of Gh1 mRNA and the secretion of GH [35]. In this study, we identified a novel circRNA, circAgtpbp1, from among 4123 screened novel circRNAs in pituitary cells from sexually mature and immature rats [34] using sequencing verification and RNase R treatment. Using the University of California, Santa Cruz (UCSC) (http://genome.ucsc.edu/, accessed on 1 July 2019) and Ensembl (http://asia.ensembl.org/index.html, accessed on 1 July 2019) genome browsers, we affirmed that circAgtpbp1 consists of exons 14–19 of the Agtpbp1 gene (in order). It mainly localizes in the cytoplasm, suggesting that it is a stable molecule involved in a ceRNA mechanism.

The expression levels of circRNAs are often stage-specific and tissue-specific [27,45]. In this study, the expression of circAgtpbp1 in sexually immature rats was found to be significantly higher than that in rats of other ages. The high expression of circAgtpbp1 in the pituitary suggests that this circRNA may play a significant role in this tissue. The expression level of circAgtpbp1 was approximately the same in GH3 cells and MMQ cells and was lower in both these cell lines than in primary rat pituitary cells. These data suggest that circAgtpbp1 exhibits tissue-specific, stage-specific and cell type-specific expression. Determination of the potential function of circAgtpbp1 in the rat pituitary will be meaningful.

GH is an important polypeptide hormone that is controlled by central and peripheral signals. GHRH plays an important role in positively regulating GH secretion [46], while GHRIH can reduce the secretion of GH [47]. In many mammalian species, GH secretion and synthesis have been found to decrease with normal aging [48,49,50]. This process of GH secretion decline can cause many diseases [51]. A lack of GH can cause dwarfism and, if left untreated, has been reported to shorten the median lifespan [52], while abnormally high levels of GH can cause acromegaly and significantly increase mortality in rodents and humans [53,54,55]. Therefore, it is important to clarify which factors affect the secretory mechanism of GH. Many studies have reported that miRNAs can affect GH secretion [35,56,57]. However, research on circRNAs that regulate GH secretion, especially circRNAs that act as molecular sponges of miRNAs, is rare. In this study, we found that Gh1 expression and GH secretion are affected by the promotion or inhibition of the expression of circAgtpbp1 in both GH3 cells and rat pituitary cells.

Previous studies have revealed that circRNAs can exert their effects in multiple ways, such as by interacting with RISC complexes to inhibit the effects of miRNAs on target genes [42,43,58]. ciRNAs and EIciRNAs can also interact with RNA polymerase II (Pol II) and U1 snRNP at the promotor of the gene in the nucleus [41]. Furthermore, circRNAs can interact with RNA-binding proteins like linear RNA to make the binding proteins active or inactive [59,60,61]. However, most circRNAs function as sponges of miRNAs, weakening the inhibitory effects of the miRNAs on their target genes. Many circRNAs have been found to affect tumorigenesis and tumor growth through their sponging functions. Thus, circRNA have become new targets for cancer research. Many circRNAs have been found to act as miRNA sponges to participate in tumorigenesis in many cancers, such as gastric cancer [62], bladder cancer [63], breast cancer [20], hepatocellular carcinoma [64], non-small cell lung cancer [65], thyroid cancer [22], and colorectal cancer [23]. CircRNAs affect various processes in tumor cells, such as cell proliferation [66], apoptosis [67], migration [68], and invasion [69]. In addition to affecting the status of tumor cells, circRNAs also regulate the biological processes of normal cells. For example, the circular RNA SNX29 regulates the proliferation and differentiation of myoblasts by sponging miR-744 [70]. CircFGFR4 binds miR-107 to promote the differentiation of myoblasts [71]. However, it has remained unclear whether circRNAs can regulate the secretion of hormones by sponging miRNAs in the pituitary. In this research, we identified a novel circRNA, named circAgtpbp1, that has a binding site for miR-543-5p, and we found that this circRNA can suppress the secretion of GH. When we overexpressed/knocked down circAgtpbp1, Gh1 expression and GH secretion levels were altered, indicating that circAgtpbp1 can reduce the inhibitory effect of miR-543-5p on Gh1. Our study proved that circAgtpbp1 could act as a ceRNA, competitively bound miR-543-5p and attenuated the inhibitory effect of miR-543-5p on its target gene *Gh1*. This may be a new method to regulate the growth of animals. Therefore, it is necessary to further clarify the regulatory mechanism of circAgtpbp1 in the pituitary.

In conclusion, we demonstrated an essential role of circAgtpbp1 in the regulatory mechanisms of GH secretion for the first time. We also elucidated the endogenous competitive relationships among circAgtpbp1, miR-543-5p and Gh1. We found that circAgtpbp1 can reduce the inhibitory effect of miR-543-5p on Gh1 and further increase the secretion of GH by sponging miR-543-5p (Figure 7). These findings indicate that circAgtpbp1 can regulate Gh1 expression in the rat pituitary at the level of Gh1 transcription. Our data contribute to the body of research on this topic, providing novel insight into the regulatory mechanisms of circRNAs.

## 5. Conclusions

Overall, the evidence generated by our study suggests that circAgtpbp1 can act as a sponge of miR-543-5p to reduce the inhibitory effect of miR-543-5p on Gh1 and further promote GH secretion. Our findings add to the existing knowledge regarding hormone regulation. 

## Figures and Tables

**Figure 1 animals-11-00558-f001:**
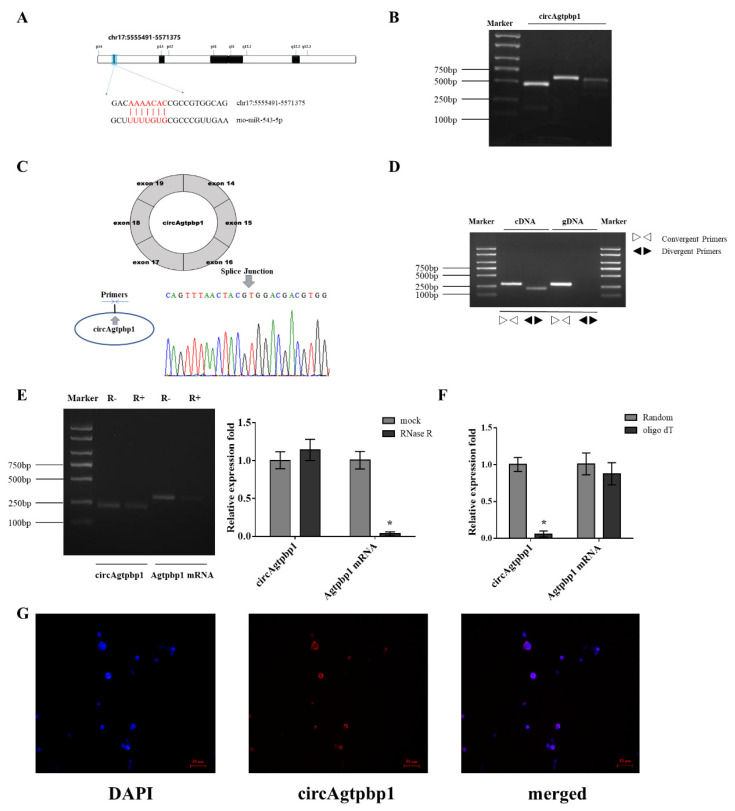
Identification and validation of circular RNAs (circRNA). (**A**) Schematic illustration demonstrates the binding sites of miR-543-5p in rno_circ_0001059. CircAgtpbp1 is produced by the Agtpbp1 gene of exons in chromosome 17. (**B**) The presence of rno_circ_0001059 was validated by PCR followed by agarose gel electrophoresis and Sanger sequencing. (**C**) Arrows represent divergent primers binding to the genomic region of circAgtpbp1. (**D**) RT–PCR assay with divergent or convergent primers indicated the existence of circAgtpbp1 in rat pituitary cells. GAPDH was used as a negative control. (**E**) RT–qPCR analysis of circAgtpbp1 and Agtpbp1 mRNA the relative expression level after treatment with RNase R. (**F**) The relative circAgtpbp1 and Agtpbp1 mRNA levels were analyzed by RT–qPCR after the reverse transcription with Random hexamer or oligo (dT)18 primers. (**G**) RNA fluorescent in situ hybridization (FISH) for circAgtpbp1. CircAgtpbp1 probes were labeled with Cy3. Nuclei were stained with DAPI. At least three replicates of each experiment were performed. Mean values and standard deviations (SD) of data are shown. One-way ANOVA was applied to determine statistical significance. *p* < 0.05 was considered significant. *, *p* < 0.05.

**Figure 2 animals-11-00558-f002:**
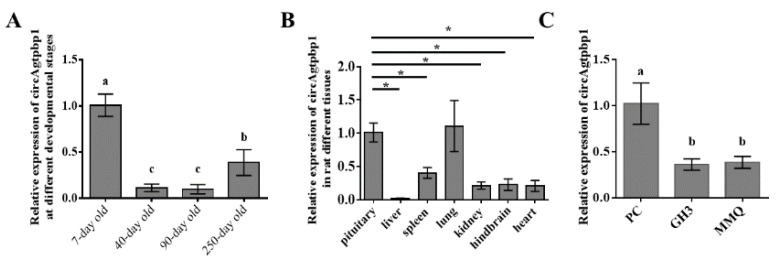
CircAgtpbp1 expression pattern in rat anterior pituitary. (**A**) RT–qPCR analysis of circAgtpbp1 the relative expression in different stages. (**B**) The relative expression level of circAgtpbp1 in mature rat different tissues. (**C**) CircAgtpbp1 expression in rat pituitary primary cell and GH3, MMQ cell lines. At least three replicates of each experiment were performed. Mean values and standard deviations (SD) of data are shown. One-way ANOVA and independent-samples *t*-test were applied to determine statistical significance. *p* < 0.05 was considered significant. Different letters (a, b and c) indicate significant differences between groups (*p* < 0.05). *, *p* < 0.05.

**Figure 3 animals-11-00558-f003:**
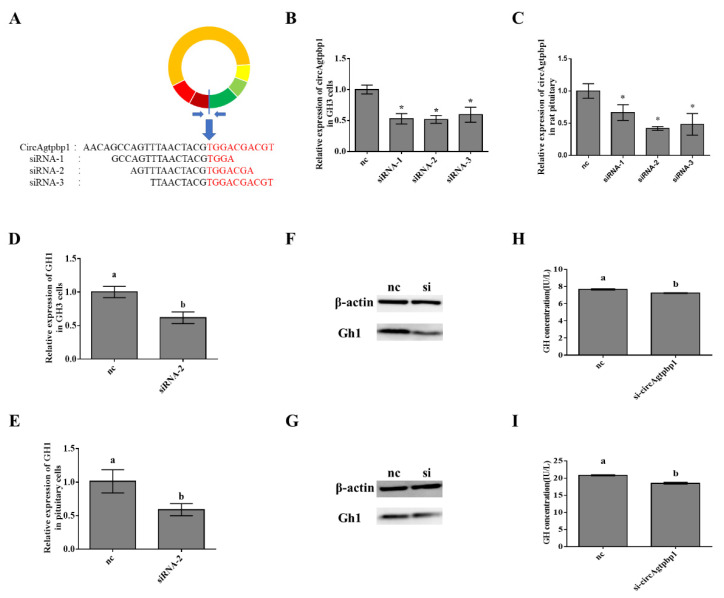
Effects of circAgtpbp1 knockdown on Gh1 transcription. (**A**) Schematic illustration showing three siRNAs that the back-splicing junction site of circAgtpbp1.Schematic illustration showing siRNAs and circAgtpbp1 expression vectors. (**B**) The expressions of circAgtpbp1 were determined with RT–qPCR in GH3 cells transfected with siRNAs. (**C**) The expressions of circAgtpbp1 were determined with RT–qPCR in pituitary cells transfected with siRNAs. (**D**) The expressions of Gh1 were determined with RT–qPCR in GH3 cells transfected with siRNA-2. (**E**) The expressions of Gh1 were determined with RT–qPCR in pituitary cells transfected with siRNA-2. (**F**) The expressions of Gh1 were determined with WB in GH3 cells transfected with siRNA-2. (**G**) The expressions of circAgtpbp1 were determined with WB in pituitary cells transfected with siRNA-2. (**H**) ELISA experiment detected GH secretion in GH3 cells after transfected with circAgtpbp1 siRNA-2. (**I**) ELISA experiment detected GH secretion in pituitary cells after transfected with circAgtpbp1 siRNA-2. At least three replicates of each experiment were performed. Mean values and standard deviations (SD) of data are shown. One-way ANOVA was applied to determine statistical significance. *p* < 0.05 was considered significant. Different letters (a,b) indicate significant differences between groups (*p* < 0.05). *, *p* < 0.05.

**Figure 4 animals-11-00558-f004:**
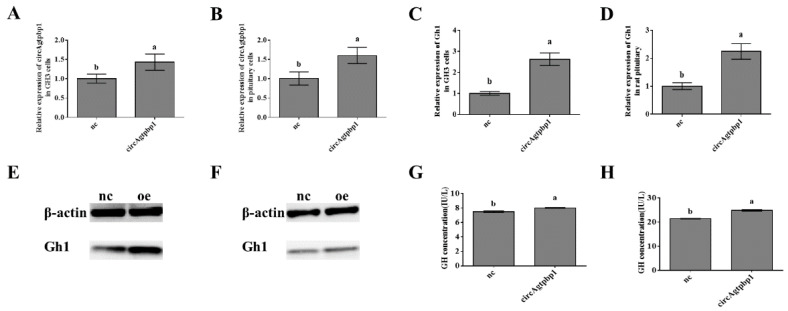
Effects of overexpressing circAgtpbp1 on Gh1 transcription. (**A**) The expressions of circAgtpbp1 were determined with RT–qPCR in GH3 cells transfected with circAgtpbp1 overexpression vector. (**B**) The expressions of circAgtpbp1 were determined with RT–qPCR in pituitary cells transfected with circAgtpbp1 overexpression vector. (**C**) The expressions of Gh1 were determined with RT–qPCR in GH3 cells transfected with circAgtpbp1 overexpression vector. (**D**) The expressions of Gh1 were determined with RT–qPCR in pituitary cells transfected with circAgtpbp1 overexpression vector. (**E**) The expressions of Gh1 were determined with WB in GH3 cells transfected with circAgtpbp1 overexpression vector. (**F**) The expressions of Gh1 were determined with WB in pituitary cells transfected with circAgtpbp1 overexpression vector. (**G**) ELISA experiment detected GH secretion in GH3 cells after transfected with circAgtpbp1 overexpression vector. (**H**) ELISA experiment detected GH secretion in pituitary cells after transfected with circAgtpbp1 overexpression vector. At least three replicates of each experiment were performed. Mean values and standard deviations (SD) of data are shown. One-way ANOVA was applied to determine statistical significance. *p* < 0.05 was considered significant. Different letters (a,b) indicate significant differences between groups (*p* < 0.05).

**Figure 5 animals-11-00558-f005:**
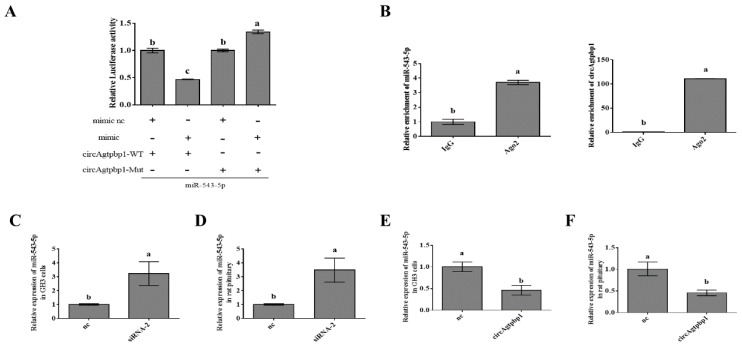
CircAgtpbp1 acts as a sponge for miR-543-5p in rat pituitary primary cells. (**A**) Luciferase assays were performed to detect the luciferase activities of 293 T cells to confirm the interaction between miR-543-5p and circAgtpbp1. (**B**) The Ago2 RIP assay showed that Ago2 significantly enriched miR-543-5p and circAgtpbp1 in rat pituitary cells. (**C**) MiR-543-5p expression increased upon circAgtpbp1 knockdown by siRNA-2 in GH3 cells. (**D**) MiR-543-5p expression increased upon circAgtpbp1 knockdown by siRNA-2 in rat pituitary cells. (**E**) MiR-543-5p expression decreased after circAgtpbp1 overexpression in GH3 cells (**F**) MiR-543-5p expression decreased after circAgtpbp1 overexpression in rat pituitary cells. At least three replicates of each experiment were performed. Mean values and standard deviations (SD) of data are shown. One-way ANOVA and independent-samples *t*-test were applied to determine statistical significance. *p* < 0.05 was considered significant. Different letters (a,b) indicate significant differences between groups (*p* < 0.05).

**Figure 6 animals-11-00558-f006:**
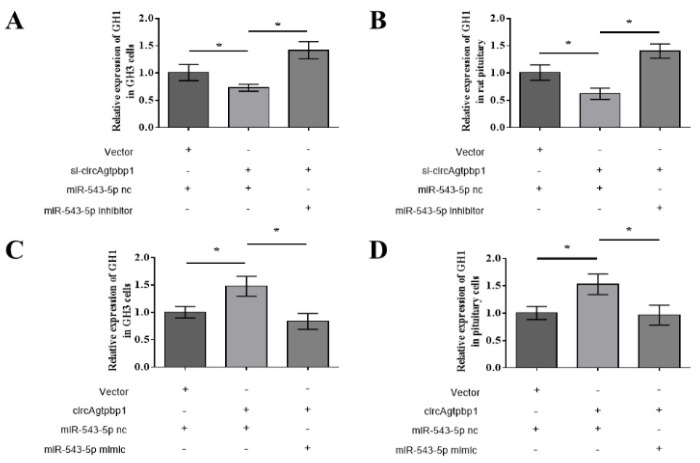
CircAgtpbp1 regulates Gh1 expression and GH secretion via targeting miR-543-5p. (**A**,**C**) The expressions of Gh1 were detected by qRT–PCR after regulating circAgtpbp1 or miR-543-5p in GH3 cells. (**B**,**D**) The expressions of Gh1 were detected by qRT–PCR after regulating circAgtpbp1 or miR-543-5p in pituitary cells. At least three replicates of each experiment were performed. Mean values and standard deviations (SD) of data are shown. One-way ANOVA and independent-samples *t*-test were applied to determine statistical significance. *p* < 0.05 was considered significant. *, *p* < 0.05.

**Figure 7 animals-11-00558-f007:**
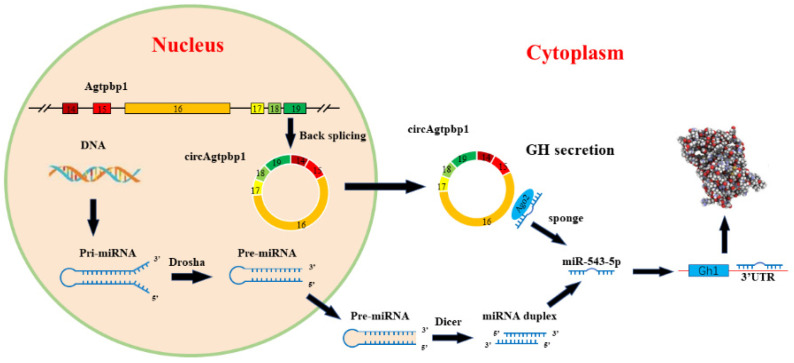
The schematic diagram shows the mechanism underlying circAgtpbp1 as a ceRNA for miR-543-5p.

## Data Availability

Data is contained within the article or Appendix A.

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
