# Peer review of "CircAgtpbp1 Acts as a Molecular Sponge of miR-543-5p to Regulate the Secretion of GH in Rat Pituitary Cells"

_animals, 2021, doi:10.3390/ani11020558_

Round 1

Reviewer 1 Report

The authors of the manuscript intended to identify the presence of CircRNAs in rat pituitary cells and study their influence on GH secretion. This interesting and up to date topic falls within the scope of the Animals. The manuscript brings very interesting data to the field. However, the standard of English claims appropriate editing and the methods used are often scarcely described. Moreover, obtained data should be further discussed

Therefore, I recommend this manuscript for publication after a major revision.

Specific recommendations

Abstract

L17 and 27 Please rephrase Our findings add to the existing knowledge regarding hormone regulation

L19 The circAgtpbp1 characteristics were authenticate through… instead of Authenticating the circAgtpbp1 characteristic through…

L24 molecule not molecular  

L61 …from introns not composed of introns

L70 Please define ceRNA

L77 Please delete “Our findings add to the existing knowledge regarding hormone regulation.”

L91-92 The authors should introduce a brief description of the methods used to obtain and culture the rat anterior pituitary cells. Did you perform cultures of anterior pituitary cells from rats at different ages (7, 40, 90 and 250 days) as referred in section results (L213)?

L136 to or for the target miRNA. Correct please.

L146-150 Please provide more information about FISH technique

L150 Please provide information about the fluorescence microscopy

L172 The methods used to identify and validate circRNA should be provided in the section Material and methods and not only in the results

L347 …are generated from exons instead of consist of exons

L352 Han, Wang et al. 2019

L383-410 The present manuscript revealed interesting data that should be further discussed in light of its impact on animal physiology

Reviewer 2 Report

Only used male rats, would be nice to actually have a balanced design between male and female.

Need to describe better the molecular biology methods. Example: cycling parameters and recipe for RT-qPCR or exactly how protein was quantified for Western blot.

Could use more evidence of circAgtpbp1 being circular.  For instance, running a gel with the circularized version of circAgtpbp1 alongside a digested linearized version.

Would also be nicer to focus on on a few cells in Fig 1G to have better resolution of the FISH results

I do like the extensive characterization across developmental stages and tissues in rats

What is the siRNA KO efficiency (%)--Fig3? Does not seem great.

Still did not drive home the "sponge" idea here...seems like circAgtpbp1 sequesters miRNA, but doesn't release it after like a sponge would (aka no releasing mechanisms characterized here)

Great usage of the molecular biology toolbox to characterize the circAgtpbp1 system!

Round 2

Reviewer 1 Report

The authors accomplished all my recommendations significantly improving the quality of the manuscript.
